# Dynamic MRI of the Mesenchymal Stem Cells Distribution during Intravenous Transplantation in a Rat Model of Ischemic Stroke

**DOI:** 10.3390/life13020288

**Published:** 2023-01-20

**Authors:** Elvira A. Cherkashova, Daria D. Namestnikova, Ilya L. Gubskiy, Veronica A. Revkova, Kirill K. Sukhinich, Pavel A. Melnikov, Maxim A. Abakumov, Galina D. Savina, Vladimir P. Chekhonin, Leonid V. Gubsky, Konstantin N. Yarygin

**Affiliations:** 1Department of Neurology, Neurosurgery and Medical Genetics, Department of Medical Nanobiotechnology, Pirogov Russian National Research Medical University of the Ministry of Healthcare of the Russian Federation, 117977 Moscow, Russia; 2Federal Center of Brain Research and Neurotechnologies of the Federal Medical Biological Agency, 117513 Moscow, Russia; 3Cell Technology Laboratory, Federal Research and Clinical Center of Specialized Medical Care and Medical Technologies of the Federal Medical Biological Agency of Russian Federation, 115682 Moscow, Russia; 4Laboratory of Cell Biology, Orekhovich Institute of Biomedical Chemistry of the Russian Academy of Sciences, 119121 Moscow, Russia; 5Department of Fundamental and Applied Neurobiology, Serbsky Federal Medical Research Centre of Psychiatry and Narcology of the Ministry of Healthcare of Russian Federation, 119034 Moscow, Russia; 6Drug Delivery Systems Laboratory, D. Mendeleev University of Chemical Technology of Russia, 125047 Moscow, Russia; 7Department of General Pathology and Pathophysiology, Russian Medical Academy of Continuous Professional Education, 125284 Moscow, Russia

**Keywords:** stroke cell therapy, intravenous transplantation, MRI, mesenchymal stem cells, MCAO

## Abstract

Systemic transplantation of mesenchymal stem cells (MSCs) is a promising approach for the treatment of ischemia-associated disorders, including stroke. However, exact mechanisms underlying its beneficial effects are still debated. In this respect, studies of the transplanted cells distribution and homing are indispensable. We proposed an MRI protocol which allowed us to estimate the dynamic distribution of single superparamagnetic iron oxide labeled MSCs in live ischemic rat brain during intravenous transplantation after the transient middle cerebral artery occlusion. Additionally, we evaluated therapeutic efficacy of cell therapy in this rat stroke model. According to the dynamic MRI data, limited numbers of MSCs accumulated diffusely in the brain vessels starting at the 7th minute from the onset of infusion, reached its maximum by 29 min, and gradually eliminated from cerebral circulation during 24 h. Despite low numbers of cells entering brain blood flow and their short-term engraftment, MSCs transplantation induced long lasting improvement of the neurological deficit, but without acceleration of the stroke volume reduction compared to the control animals during 14 post-transplantation days. Taken together, these findings indicate that MSCs convey their positive action by triggering certain paracrine mechanisms or cell–cell interactions or invoking direct long-lasting effects on brain vessels.

## 1. Introduction

Stem cell-based therapy is the promising strategy for the treatment of many neurological disorders, including ischemic stroke [1,2,3]. With regard to safety and bioethical issues, mesenchymal stem cells (MSCs) are among the most relevant candidates for translational research and implementation into clinical practice [4]. Transplantation of MSCs relieved neurological impairment in experimental animal models of cerebral ischemia [5,6,7,8,9]. Transplantation into humans in clinical trials proved to be safe and feasible and showed promising results [10,11,12]. Taken together, the already conducted studies have shown that MSCs have a unique set of properties, including low immunogenicity, anti-inflammatory, immunomodulatory, and neuroprotective activity, the ability to stimulate angio- and neurogenesis, to transfer functional mitochondria to the damaged cells, and to promote repair of the blood brain barrier [6,10,13,14,15,16,17]. However, despite rigorous studies, the precise mechanisms of MSCs’ action are not fully understood.

Positive therapeutic effects were demonstrated after various ways of cell transplantation (intra-arterial, intravenous, intracerebral, and others), but the optimal one has not been established yet [6,18,19]. Each delivery route has its advantages, constraints, and suits certain needs [20]. In case of MSCs transplantation in the acute stroke, systemic (intra-arterial and intravenous) administration seems to be the most appropriate. The intra-arterial way of stem cell transplantation has shown efficacy and provides the delivery of a large number of cells directly into the brain vessels, bypassing filtering organs [21,22]. This route of administration has become even more relevant due to the widespread use of endovascular treatment methods in acute stroke patients [23], despite this procedure requiring technical precision due to the risk of cerebral embolism [24]. Intravenous administration (IV) has also shown therapeutic efficacy and is the most commonly used, because it is less invasive, has minimal risk of adverse events, can be easily applied multiple times, and is suitable even for patients in critical clinical conditions [20,25,26,27]. Finding the optimal route of transplantation, as well as timing and the dosage of cells, is a challenging task due to the poor understanding of the exact mechanisms of MSCs’ action. Further investigations are needed in order to accelerate the translation of this approach to clinical practice. In this way, the studying of cell distribution, homing, and fate after transplantation seems to be one of the most important tasks [28].

Among methods of the in vivo cell tracking, magnetic resonance imaging (MRI) is one of the most widely used in experimental studies and clinical trials, since it is convenient, non-invasive, and provides high spatial resolution of brain tissues [29,30,31]. In order to visualize transplanted stem cells, different T1 or T2 contrast agents were developed [32]. T1 contrast agents are mainly based on gadolinium (Gd)-containing compounds and are widely utilized in clinical practice due to high biocompatibility. However, Gd-based contrast agents are relatively rarely employed for stem cell labeling because of their limited ability to accumulate within the cells and the lack of knowledge about their effects on cellular functions [33,34,35,36]. T2 contrast agents are based on the paramagnetic nanoparticles (size from tens of nm to 1 μm) [34]. Among them, superparamagnetic iron oxide (SPIO) particles have been successfully employed for stem cells in vivo tracking for more than two decades, and during this time, various types of SPIOs were invented and their properties were thoroughly investigated [29,37,38,39,40]. SPIOs are considered to be good contrast agents in MRI due to their high loading capacity in stem cells and the ability to short T2 and T2* relaxation time [35,41,42]. For the best visualization of SPIO labeled stem cells, T2* weighted images (T2* WI) and relatively new more sensitive susceptibility weighted images (SWI) are ordinarily used [43,44]. SPIO-labeled cells are detected as hypointense spots on the MR images because SPIO particles create a local inhomogeneity of the magnetic field and decrease T2* relaxation time in the surrounding regions (well reviewed by [34]). It has been demonstrated that SPIO-labeled stem cells can be detected at a single-cell level due to the development of advanced MRI protocols [44,45,46,47,48].

For a long time, the MR tracking of SPIO-labeled stem cells has been carried out after the completion of cell injection [35]. However, the tracing of transplanted cells right at the moment of infusion is also important. The monitoring of cell homing in real-time may help to better understand the mechanisms of cells’ action depending on the route of administration. Moreover, it provides an opportunity for the immediate detection of insufficient or incorrect biodistribution related to the imperfect injection of stem cells, thus improving the overall therapy efficiency. Due to the development of fast MRI protocols, dynamic tracking of stem cells right at the moment of transplantation has become possible using dynamic (also called real-time or time-lapse) MRI [24,49,50,51].

According to the available data, real-time tracing of MSCs in the brain was described only for the intra-arterial (IA) delivery [24,50,51], but there are no reports about cells’ distribution in the brain during intravenous injection. At the same time, the overall biodistribution in the whole body of the intravenously injected MSCs has been well described and it was shown that transplanted cells reach the brain in just tiny numbers [6,19,52,53]. In some in vivo studies, MSCs were even not detected in the brain after IV transplantation, probably due to the low cell dosage or/and wrong cell visualization method [19,54]. The existing MR protocols used for tracing cells after the intra-arterial cell delivery may be not sensitive enough to detect the small numbers of cells reaching the brain after the intravenous injection.

The main goal of this article was to study the dynamic distribution of MSCs in the ischemic brain during and after IV transplantation. To perform this, 2 × 10^6^ of the SPIO-labeled MSCs were intravenously injected into rats directly inside the MR scanner 24 h after the transient middle cerebral occlusion. Transplanted cells were traced for one hour from the beginning of cell infusion and 24 h later. To enable precise detection of single transplanted MSCs in the rat brain, we changed the MR protocol previously used by us for dynamic cell detection after intra-arterial transplantation [51]. The obtained results demonstrated the possibility of dynamic stem cell tracking by MRI during intravenous MSCs administration and provided the pattern of distribution and homing of transplanted cells in the ischemic rat brain. Additionally, we aimed to evaluate the therapeutic efficacy of MSCs after their IV administration in the chosen dosage in the experimental ischemic stroke model.

## 2. Materials and Methods

### 2.1. Animals

All procedures with laboratory animals were performed in accordance with the guidelines of the Declaration of Helsinki and directive 2010/63/EU on the protection of animals used for scientific purposes of the European Parliament and the Council of European Union dated 22 September 2010 and were approved by the Pirogov Russian National Research Medical University Animal Care and Use Commission (protocol code No 24/2021 from 10 December 2021). In vivo studies are reported according to the ARRIVE guidelines. Male Wistar rats (weighing 250–300 g, purchased from “SMK STESAR”, Vladimir, Russia, LLC) were used for the experiment. Males were chosen to avoid the potential neuroprotective effects of estrogens [55]. The animals were housed in groups of four to five animals per cage before surgery. They were kept under 12-h/12-h light/dark cycle, room temperature 22 ± 2 °C, humidity 45–65%, and free access to standard rodent chow and water.

### 2.2. Cell Culture

Human mesenchymal stem cells were isolated from the donor’s placenta collected after normal delivery at 38–40 weeks of gestation. The biological material was obtained from the Perinatal Center of Kama Children’s Medical Center of Naberezhnye Chelny after getting informed consent from mothers using a conventional procedure described previously [56]. The human placenta tissue samples were grinded by blade, extensively washed with Dulbecco’s phosphate-buffered saline (DPBS) and then incubated for 2 h at 37 °C in DPBS containing 1 mg/mL of collagenase I type (Gibco, Thermo Fisher Scientific, Waltham, MA, USA). After incubation, the collagenase was inactivated by 10% fetal bovine serum (FBS) in DPBS. For the cell sedimentation, the tissue suspension was centrifuged at 400× *g* for 4 min at 25 °C. The supernatant was discarded and tissue pieces resuspended in the complete culture medium comprising DMEM-F12, 2 mM L-glutamine, 100 U/mL penicillin, 0.1 mg/mL streptomycin, and 10% FBS (all reagents from Gibco, Thermo Fisher Scientific, Waltham, MA, USA). The resuspended tissue pieces were placed in T 75 culture flasks and maintained in a humidified atmosphere under standard conditions (37 °C, 5% CO_2_). Cells were allowed to adhere for 3 days and non-adherent cells were removed by replacing the medium. Upon reaching 80% confluence, adherent pMSCs were harvested by trypsinization and subcultured at 1:3 ratio in T 75 flasks. Cells were cultured until they reached 80% confluence in each passage and collected for transplantation after 3–5 passages. The quality of the obtained cells was confirmed by flow cytometry of major MSC’s markers as CD34−, CD45−, HLA−DR−, CD105+, CD29+, CD73+, CD90+. The absence of mycoplasma contamination was confirmed by PCR. One day before injection, the pMSC were labeled with superparamagnetic iron oxide (SPIO) microparticles (MC03F Bangs Laboratories, USA, mean diameter 0.50 ± 0.99 μm) carrying the Dragon Green fluorescent dye (λex = 480 nm, λem = 520 nm) according to the manufacturing protocol as previously described [44]. On the next day before, transplantation cells were dissociated using accutase (StemCells Technology, Vancouver, BC, Canada). After dissociation, accutase was inactivated with full media, contained 10% FBS, and the cell suspension centrifuged at 400× *g* for 4 min. After that, the cells were also labeled with lipophilic membrane red fluorescent dye PKH26 (Sigma-Aldrich, Burlington, MA, USA) as previously described [44]. Prior to transplantation, cells were washed twice with DPBS and counted. A dose of 2 × 10^6^ cells in 1 mL of saline was prepared for each intravenous transplantation.

### 2.3. Study Design

All rats (n = 50) were subjected to the transient 90 min middle cerebral artery occlusion. Then, 24 hours after the operation, MRI (T2 WI, Diffusion Weighted Imaging—DWI) was performed for the confirmation of ischemic zone formation (8 rats were excluded from the experiment due to unsuccessful stroke modeling). Some of the rats (n = 12) were included into the study of cell distribution and received IV infusion of the SPIO labeled MSCs directly inside the MR scanner (n = 12). In order to evaluate the dynamic distribution of transplanted cells, the high resolution SWI were performed every 7 min 15 s within 58 min. After the end of the MR examination, the histological study was performed. A schematic image of the study design is presented in Figure 1.

The leftover animals (n = 30) were included in the study to evaluate the therapeutic effects of cell transplantation. After ischemic stroke, modeling rats were randomly divided into two groups: (group 1) animals with IV transplantation of 2 × 10^6^ MSCs (n = 15); (group 2) rats with IV transplantation of 1 ml of saline (n = 15). The dynamics of the reduction of neurological deficit and of stroke volume were evaluated right before the IV administration and on the 7th and 14th day after the infusion of MSCs (or saline in the control group).

### 2.4. Transient Middle Cerebral Artery Occlusion

The surgery was performed under the isoflurane inhalation anesthesia (Aerrane, Baxter HealthCare Corporation, Deerfield, IL, USA) using an animal anesthesia system (E-Z-7000 Classic System, E-Z-Anesthesia^®^ Systems, Palmer, PA, USA). Additionally, a subcutaneous injection of 0.1–0.2 mL 0.5% bupivacaine into the surgery field and an intraperitoneal premedication with atropine sulfate 0.05 mg/kg in 1 mL 0.9% NaCl were performed. For the induction of anesthesia, 3.5–4% isoflurane mix with atmospheric air was used and for the maintenance of anesthesia the percentage of isoflurane was reduced to 2–2.5%. Transient 90 minutes middle cerebral artery occlusion (tMCAO) with MRI guiding was performed as described previously [57]. In brief, the bifurcation of the right common carotid artery was exposed. The monofilament with a rubber-coated tip (Doccol Corporation, Sharon, MA, USA, diameter 0.19 mm, length 30 mm; diameter with coating 0.37 ± 0.02 mm; coating length 3–4 mm) was inserted through the external carotid artery into the lumen of the internal carotid artery till the origin of the middle cerebral artery. The occlusion period lasted for 90 min, after which the monofilament was removed and the surgical wound was sutured, 3 mL of sterile saline was injected intraperitoneally and 30 mg/kg gentamicin sulfate was given intramuscularly. During recovery from anesthesia, the operated rats were placed in preheated cages. Rats with unsuccessful stroke modeling were excluded from the study after the MRI was performed 24 hours after operation (n = 8).

### 2.5. Intravenous Transplantation

Intravenous administration of MSCs was performed via the right femoral vein 24 h after experimental stroke modeling. The procedure was carried out under inhalation anesthesia, as described above, despite that for intravenous transplantation, the isoflurane was mixed with pure oxygen in order to increase the concentration of oxyhemoglobin in the venous blood and attenuate the visibility of cerebral veins to improve the MR detection of SPIO labeled cells on the SWI. After subcutaneous injection of 0.1–0.2 mL 0.5% bupivacaine into the surgery field of lower limb and intraperitoneal premedication with atropine sulfate 0.05 mg/kg in 1 mL 0.9% NaCl, the femoral vein was isolated and catheterized with a microcatheter (Doccol Corporation, Sharon, MA, USA, rodent tail vein catheter with diameter 1F, 120 cm) filled with saline and connected to the syringe filled with 2 × 10^6^ cells in 1 mL of saline. Then, the animal was carefully transported to the MR scanner. MSCs transplantation was performed using the nanoinjector (Leica Microsystems GmbH, Wetzlar, Germany) at a rate of 250 μL/min for 4 min. At the end of the infusion and MR examination, the rat was transferred back to the operation room, where the microcatheter was removed, the surgical wound was closed with sutures and 30 mg/kg gentamicin sulfate was given intramuscularly.

### 2.6. Estimation of Neurological Deficit

For estimation of the neurological deficit of experimental rats the modified neurological severity scale (mNSS) was used [58]. The mNSS was designed for rodents and allows to perform the complete assessment of neurological status: motor and sensory impairment, as well as balance and reflex disorders. The evaluation was carried out right before the IV administration and on the 7th and 14th day after the infusion. All tests were conducted by the observers blinded with regard to the treatment groups.

### 2.7. Magnetic Resonance Imaging

All MR examinations were carried out using a 7T ClinScan system for small animals (Bruker BioSpin, Billerica, MA, USA). During the procedure, rats were maintained under inhalation anesthesia with 3.5–4% isoflurane mixed with atmospheric air for MRI guided MCAO surgery (previously described in [57]) and mixed with pure oxygen for other procedures. Diffusion-weighted imaging (DWI) with mapping of the apparent diffusion coefficient (ADC) (Echo-planar pulse sequence; TR/TE = 5000/22 ms; 14 b-factors from 0 to 2000 s/mm^2^; diffusion directions = 6; averages = 1; spectral fat saturation; FOV = 32 mm × 20 mm; slice thickness = 1.0 mm; matrix size = 80 × 52) and T2-weighted imaging (T2 WI, Turbo Spin Echo pulse sequence with restore magnetization pulse; turbo factor = 9; TR/TE = 6000/46 ms; averages = 1; spectral fat saturation; FOV = 30 × 21 mm; slice thickness = 0.7 mm; matrix size = 256 × 162; respiratory gated) were obtained for evaluation of the infarct area before MSCs injection. For the visualization of the SPIO-labeled MSCs in the rat brain, repeated high resolution susceptibility weighted imaging (SWI) was obtained with the following parameters: 3D Gradient Echo with RF spoiling and flow compensation; TR/TE = 50/19.1 ms; flip angle = 15; averages = 1; spectral fat saturation; FOV = 19 mm × 19 mm; slice thickness = 0.5 mm; matrix size = 192 × 192) with temporal resolution 7 min 15 s and 8 repeats within 58 min (approximately 1 h) after the start of IV injection. Additionally, the same high resolution SWI was performed 24 h after the IV transplantation of MSCs. High resolution SWI has spatial resolution of 0.1 mm × 0.1 mm × 0.5 mm. The distribution of the SPIO-labeled cells was evaluated manually by analyzing dynamic high resolution SWI. Labeled cells were defined as hypointense spots appearing on the images after IV injection in four areas: infarct core, peripheral to infarct core zone, contralateral hemisphere, and brainstem.

### 2.8. Estimation of Stroke Volume

MRI data analysis was performed as described previously [59] using ImageJ software (Rasband, W.S., ImageJ, U. S. National Institutes of Health, Bethesda, MD, USA, 1997–2015). The volume of the infarct area on T2 WI was measured manually by the same person (radiologist) and calculated according to the formula: V = (S1 + … + Sn) × (h + d), where S1, …, Sn is the infarct zone measured on slice n, where h is the thickness of the slice and d is the interval between the slices.

### 2.9. Histology

For histochemical studies, animals were sacrificed at 1 h (n = 9) and 24 h (n = 3) after IV transplantation of MSCs. The rats were euthanized by inhalation anesthesia with a lethal dose of isoflurane and additional injection of a lethal dose of Zoletil. Then, transcardial perfusion was performed using 4% paraformaldehyde (PFA) in 0.1M PBS. The brains and additionally lungs, liver, and kidneys were harvested and postfixed at 4 °C overnight in the PFA, washed three times with PBS, and cryoprotected in 30% sucrose solution. Then, 40 um coronal sections were cut using a cryostat (Leica CM1900, Nussloch, Germany). The sections with the Dragon green and/or PKH 26 fluorescence were collected and stained with DAPI solution (2 µg/mL, Sigma-Aldrich, Burlington, MA, USA). Then, sections were mounted in 80% glycerol. Fluorescence images were captured with the Nikon A1R MP + laser scanning confocal microscope (Nikon Instruments Inc., Melville, NY, USA).

### 2.10. Statistical Analysis

Statistical analysis was performed using IBM SPSS Statistics 23 (IBM Corp., Armonk, NY, USA). The evaluation of the dynamics of changes of mNSS and stroke volume was performed by the comparisons between therapeutic and control groups using the General Linear Model with repeated measurements (values obtained on day 7 and 14 were normalized based on data captured on day 1). Additionally, for evaluation of the differences in mNSS score between groups on days 7 and 14, Mann–Whitney U test with Bonferroni correction was performed. *p*-value < 0.05 indicated differences with statistical significance. Descriptive statistics were used for estimation of SPIO-labeled MSCs distribution in the brain during their intravenous transplantation. After visual assessment of the high resolution SWI, the diagrams were generated.

## 3. Results

### 3.1. Dynamic MRI Distribution of MSCs in Ischemic Rat Brain after Intravenous Transplantation

Dynamic MR visualization of SPIO-labeled MSCs in the rat brain was started at the beginning of intravenous infusion and dynamic visualization was continued for 1 h. To do this, cell transplantation was carried out inside the MR scanner and high resolution SWI was performed approximately every 7 min. The example of the obtained images is shown in Figure 2. SPIO-labeled MSCs were visualized as distinct single hypointense spots on SWI due to creation of the disturbance of the local magnetic field and the reduction of T2* relaxation time by the SPIO label [44] (marked with red arrows on Figure 2). It is important to note that accumulation of SPIO labels within the cells create disturbance of the local magnetic field of the surrounding tissue within a volume that exceed the real volume occupied by the injected MSCs (average MSCs diameter 24–25 μm). This feature is known as the “blooming effect”, characterized by nonlinear increase of the hypointense area with increasing of iron content [44]. In order to assess areas of labeled cells accumulation and to distinguish them from the other causes of signal loss on SWI (cerebral veins, microbleeds, and others [60]), the series of adjacent MR-slices were analyzed in dynamics. SPIO-labeled cells were distinguished from the other hypointense spots on SWI by analyzing their shape and time of appearance.

To improve the quality of cells’ visualization and for better distinction between the images of cells and cerebral vessels, the mixture of isoflurane and oxygen instead of isoflurane and air was used for inhalation anesthesia. This approach provides an increase of oxyhemoglobin and decrease of deoxyhemoglobin concentration in the venous blood. Since oxyhemoglobin is a diamagnetic and deoxyhemoglobin a paramagnetic [61], the accumulation of oxyhemoglobin in the blood reduces the visibility of cerebral veins, as shown in Figure 3.

The dynamic distribution of labeled MSCs during their intravenous transplantation and its changes within 1 h were analyzed on the basis of MRI data and are shown in Figure 4. Only single or small groups of transplanted cells were detected in the infarct and peripheral zone, contralateral hemisphere, and brainstem starting from the 7th min after the start of infusion. The number of the hypointense spots on SWI reached its maximum by 29 min in all described brain regions except the peri-infarct zone, where the maximum number of labeled cells was detected 1 h after cell administration. After 1 h, the number of cells in the contralateral hemisphere and brainstem decreased, though in the infarct and peri-infarct zones it remained stable. The obtained results demonstrate that intravenously transplanted MSCs started to enter brain vessels relatively early after entering the systemic circulation. Transplanted cells continued to accumulate in cerebral vessels within an hour and were later eliminated from cerebral circulation by 1 day.

MRI data were verified by histological examination. Prior to the transplantation, MSCs were double labeled with SPIO particles carrying the green, fluorescent dye Dragon green and with the lipophilic membrane red fluorescent dye PKH26. In full agreement with the MRI results, the histological examination detected single transplanted cells in the brain vessels of some rats immediately after the end of the dynamic MR study (Figure 5A–C). Though MRI revealed single labeled cells In all experimental animals, in the histological study, transplanted cells were found only in 3 animals from the whole experimental group. Probably, this phenomenon can be explained by partial or sometimes total removal of the transplanted cells from the cerebral vessels during transcardial perfusion necessary for the brain tissue fixation. However, many double labeled cells were visualized in the lungs (see Figure 5D), and the Dragon green label was found in the liver (Figure 5E) and kidneys (Figure 5F).

In our study, MSCs remained in the brain vessels not more than 24 h after intravenous transplantation. At 24 h, labeled cells were no longer detected in the brain, according to the results of both MR and histological examination. Data shown in Figure 6.

### 3.2. Evaluation of the Therapeutic Effects of MSCs after Intravenous Transplantation in Experimental Ischemic Stroke

The assessment of neurological deficits of rats with stroke modeling was performed using mNSS right before IV injection of cells or saline (in control group) and on the 7th and 14th day post-administration. The dynamics of changes of the neurological status within 14 days in rats from MSCs group differed significantly from that of the control group. In the cell therapy group, the mNSS score was significantly lower already 7 days after transplantation. Moreover, the pairwise comparisons between group at day 7 and 14 have also shown that mNSS score was significantly lower in the group with cell therapy. The results are shown in Figure 7.

The stroke volume was measured based on the MRI data (hyperintense zones on T2 WI) at days 1, 7, and 14 of the observation period. No significant differences in the dynamics of stroke volume reduction between the MSCs group and the control group were revealed. Results are shown in Figure 8.

## 4. Discussion

The present study was dedicated mainly to the dynamic MR visualization of the intravenously transplanted SPIO labeled MSCs in the ischemic rat brain. We aimed to estimate cell distribution both during the infusion time and the subsequent passage through the systemic and pulmonary circulation within an hour. The achievement of this goal has become possible due to the recent development of dynamical MRI for stem cell tracing. The first attempts to visualize the distribution of stem cells in dynamics at early time points after transplantation were made in 2012 [49]. The authors infused glial precursor cells intra-arterially in rats with modeled global brain inflammation and subsequently performed T2* WI 1, 10, 20, and 30 min after cell injection. The application of this technology allowed the researchers to monitor glial precursor cells at the described time intervals and to obtain quite new data about their homing to inflamed endothelium. The next step towards more precise stem cell imaging in dynamic right at the moment of cell infusion was made by Walczak et al. [24,50], who proposed cell transplantation directly inside the MR scanner and to obtain high speed T2* WI with 2 s temporal resolution simultaneously with cell administration. It has been shown that such parameters (fast MRI protocols) enabled visualization of cell homing in dynamic and confirmed the possibility of immediate intervention in the case of undesired cell distribution. Recently, we also reported the dynamic MRI of the intra-arterially transplanted neural precursor and mesenchymal stem cells using the T2* gradient echo pulse sequence with time resolution of 60 s in order to achieve better image quality of the cells necessary for the accurate cell tracking [51]. This approach allowed us to demonstrate the pattern of dynamical distribution and subsequent homing of intra-arterially transplanted stem/progenitor cells in the rat brain after experimental stroke modeling. The use of the described MR protocol with high temporal resolution and relatively low spatial resolution was optimal for the dynamical detection of the SPIO-labeled cells after IA transplantation, since in this case, high numbers of stem cells were delivered directly to the brain arteries.

The detection of MSCs after IV injection requires more sensitive protocols due to the much lower inflow of transplanted cells in the brain in comparison with IA cell administration [19]. For this reason, in the current study, we performed high resolution SWI with an increase of time scanning up to 7 min instead of fast T2* WI. It is important to note that for inhalation, anesthesia isoflurane was mixed with pure oxygen. This technical feature improved the visualization of SPIO-labeled cells due to the reduction of the MR signal from the cerebral veins. This protocol enabled us to perform accurate detection of small groups of cells and even single cells passing through the brain vessels. According to the available literature, the dynamic MRI distribution of intravenously transplanted stem cells has not been described previously, however a similar approach was successfully performed for tracing of immune cells during IV administration. Masthoff et al. [62] showed that application of dynamical (also called time-lapse) MRI with high resolution T2* WI and prolonged scan time up to 8 min allowed to observe dynamical distribution of SPIO labeled monocytes at a single-cell level. Moreover, the same research group showed that the use of dynamical MRI provided not only single cell detection, but also the monitoring of their circulation in the brain after IV injection [63]. In the present study, the use of the high-resolution dynamic MRI also allowed us to detect single MSCs in the brain and to reveal the pattern of cells’ distribution after intravenous transplantation.

The obtained results have shown the time-dependent accumulation of the intravenously injected SPIO-labeled MSCs. The maximum accumulation of cells was observed 29 min after the injection. According to the MRI and histological data, MSCs were detected in small numbers, but in all brain regions. After 1 h, the number of cells started to decrease in the contralateral (to the ischemic lesion) hemisphere and in the brainstem, but not in the infarct core and the peri-infarct zone. These findings are in line with the well known data about the ability of MSCs to amass and home to the injured areas [6,64,65]. Despite the extensive investigations of the MSCs’ mechanisms of homing and migration in vitro (reviewed by [5,10]), these processes in vivo are not fully understood yet. The previous studies also demonstrated that after the completed IV transplantation, cells were visualized in the brain in very small numbers or were not detected at all [19,53]. It is believed to be partly due to the initial trapping of transplanted cells in the lungs [66].

In the current study, we relied on the dynamical MRI and focused on the evaluation of the pattern of MSCs’ distribution in the brain only. For the investigation of the biodistribution of stem cells in the whole body, other methods of the in vivo cell detection, such as bioluminescence, radionuclide techniques, or PCR with the following histological confirmation, are more accurate and preferable. Many studies using these methods and devoted to this issue have been performed (extensively reviewed in [19,53,67]). All conducted studies have demonstrated that after IV transplantation, the vast majority of MSCs were retained in the lungs and could be detected in limited numbers in liver, spleen, kidney, and other organs [68,69,70]. Some authors supposed that the differences between the diameters of MSCs and lung capillaries, and the adhesion abilities of MSCs are the most likely reasons of the pulmonary cell trapping [71]. In the current work, we additionally performed the histological examination of lungs and other parenchymal organs 1 h after cell infusion. We found a great number of double labeled MSCs in the lungs, which is in full accordance with the previous studies [66,68,69,70,72]. However, in other organs only SPIO/Dragon green microparticles were visualized. This phenomenon requires further investigation. At the moment, we hypothesize that MSCs may be destroyed and/or phagocytized by other cells. Interestingly, recent findings indicate that positive therapeutic effects of MSCs may be mediated by phagocyting of transplanted cells by monocytes in the lungs and following redistribution of these monocytes with changed immunophenotype to other organs [73].

In our study, we did not observe long-term engraftment of human MSCs in the rat brain. After 24 h, no transplanted cells were detected in the cerebral vessels. Despite the short-term engraftment, we showed the prolonged improvement of neurological deficit of experimental animals during the observation period (14 days). Obtained results are in full accordance with the extensive literature data, which also demonstrated significant improvement of neurological impairment after IV MSCs administration in case of experimental stroke [6,26,74,75]. In the present study, we qualitatively assessed the presence or absence of the transplanted MSCs in the different brain regions in dynamics. However, the quantitative assessment of the exact number of transplanted cells and its correlation with neuro-logical deficit could be the promising direction for future research. For this purpose, the use of bioluminescence or radionuclide techniques seems to be more preferable and may provide new data about MSCs’ mechanisms of action.

In the current study, we did not see any impact of the IV MSC administration on the rate of the reduction of stroke volume. The literature data about the effects of MSCs transplantation on the infarct volume vary greatly. Some researchers reported prominent reduction of the ischemic lesion volume after MSCs transplantation [76,77,78,79]. At the same time, in many studies, the authors did not observe the significant decrease of ischemic zone size in animals with MSCs therapy comparing to the control group [51,80,81]. Interestingly, Zhang et al. have demonstrated that MSCs had no impact of stroke volume changes regardless of the route of administration [82]. The reasons for such controversial data are not clear, however it is obvious that post-stroke recovery involves not only the reduction of infarct volume.

Taken together, the revealed short-term engraftment of MSCs in the brain together with their prolonged therapeutic efficacy may indicate that transplanted cells mediate their positive action by paracrine mechanisms, as well as possibly through cell–cell interactions and trigger mechanisms exerted in the brain vessels. However, the exact mechanism requires further investigations.

## 5. Conclusions

Visualization of the intravenously transplanted MSCs in the rat brain requires high sensitivity of the detection method due to the low numbers of cells reaching the brain vessels. The SWI-based MRI protocol in combination with isoflurane/oxygen anesthesia allows the visualization of MSCs labeled with the iron oxide nanoparticles in the rat brain with the 7 min 15 s temporal resolution. The increasing accumulation of stem cells was observed during the first 28 min in the infarct and the peri-infarct zones, contralateral hemisphere, and brainstem and was followed by the decrease of their numbers at 1 h post injection and later. Transplanted cells were transiently homed in the brain vessel up to 1 day. This study shows that the SWI-based MRI protocol can be a powerful tool in the evaluation of stem cell distribution and homing after intravenous administration and can be successfully applied in the personalized cell therapy translational studies.

## Figures and Tables

**Figure 1 life-13-00288-f001:**
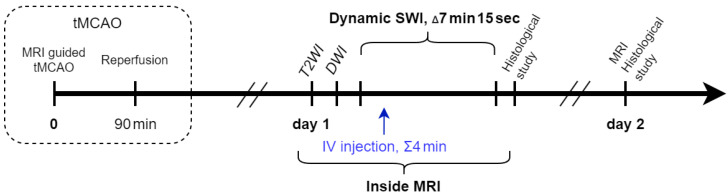
Study design. The steps of the study are schematically presented as a timeline. DWI—diffusion-weighted imaging, IV—intravenous, MRI—magnetic resonance imaging, SWI—susceptibility weighted imaging, T2 WI—T2-weighted imaging, tMCAO—temporal middle cerebral artery occlusion.

**Figure 2 life-13-00288-f002:**
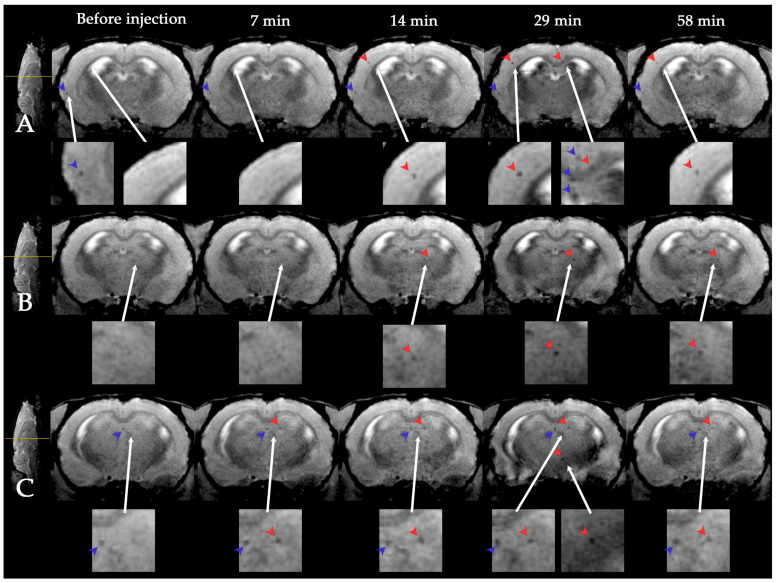
Dynamic MRI of rat brain with experimental stroke before and during 1 h after intravenous SPIO-labeled MSC transplantation. SWI of three adjacent coronal brain slices (**A**–**C**) of the same rat are presented (slicing levels marked with the yellow lines on the left column). White arrows indicate inserts with high magnification. Hypointense (dark on SWI) spots correspond to areas of SPIO labeled MSCs accumulation are marked with red arrow heads. Single cells were visualized in the infarct zone, contralateral hemisphere and brainstem starting from the 7th minute after the beginning of cell infusion. The other causes of signal loss on SWI (cerebral blood vessels) are marked with blue arrow heads. SPIO-labeled cells were distinguished from the others hypointense spots on SWI by analyzing their shape and time of appearance.

**Figure 3 life-13-00288-f003:**
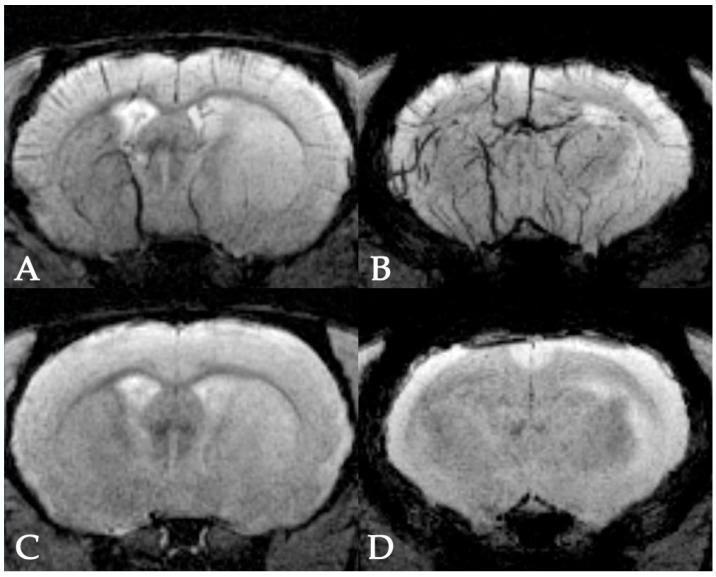
MRI of rat brain after stroke modeling under isoflurane inhalation anesthesia using atmospheric air or oxygen as the carrier gas. (**A**,**B**)—SWI of two coronal brain slices of a rat anesthetized with the mixture of isoflurane and atmospheric air. Hypointense zones and lines correspond to cerebral veins. (**C**,**D**)—SWI of two coronal brain slices of a rat anesthetized with the mixture of isoflurane and pure oxygen. Clearly, the increase of the oxyhemoglobin concentration in the venous blood attenuates the visibility of veins on MR images.

**Figure 4 life-13-00288-f004:**
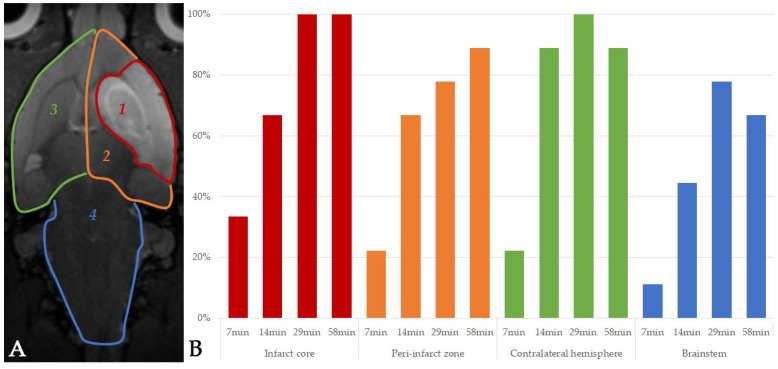
Diagrams of dynamical distribution of MSCs in the rat brain with experimental stroke within 1 h after intravenous transplantation. (**A**) The four rat brain regions where the distribution of MSCs was evaluated are marked on a T2 WI: 1—infarct core; 2—peri-infarct zone; 3—contralateral hemisphere; 4—brainstem. (**B**) The diagrams represent the percentage of animals in which single hypointense spots were visualized by SWI in different zones of the brain at the following time points: 7, 14, 29, 58 min. The colors on the diagram correspond to the colors of the borders of the four marked brain regions: infarct core, peri-infarct zone, contralateral hemisphere, and brainstem.

**Figure 5 life-13-00288-f005:**
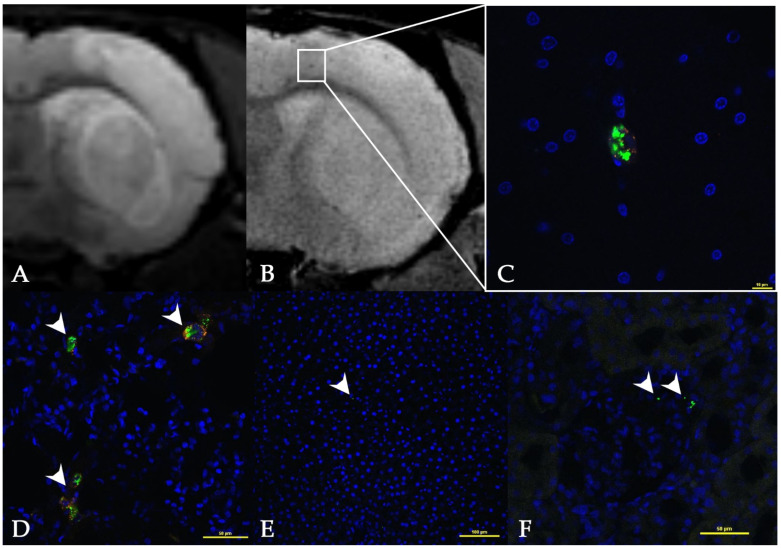
MRI and histological images of the rat brain with experimental ischemic stroke 1 h after intravenous transplantation of MSCs. (**A**) T2 WI: the hyperintense area corresponds to the ischemic lesion. (**B**) SWI: the SPIO-labeled MSCs are detected as single hypointense spots. (**C**–**F**) Confocal fluorescence images. SPIO/Dragon green labels are marked with white arrows. (**C**) Enlarged image of the area delineated as a rectangle on (**B**). A MSC double labeled with the membrane lipophilic dye PKH26 (orange) and the SPIO/Dragon green microparticles distributed in the cytoplasm (green) is clearly visualized. The nuclei were stained with DAPI (blue). Scale bar: 10 μm. (**D**) A group of double labeled MSCs detected in the lung tissue. Scale bar: 50 μm. (**E**) SPIO/Dragon green labels are detected in the liver. Scale bar: 100 μm. (**F**) SPIO/Dragon green labels visualized in the kidney. Scale bar: 50 μm.

**Figure 6 life-13-00288-f006:**
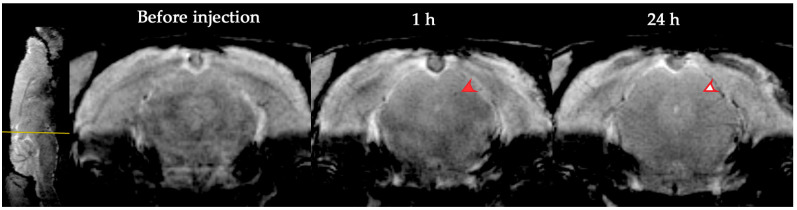
MR images of the rat brain before, 1 and 24 h after cell transplantation. On the SWI pictures of the brain taken at the same level, SPIO-labeled MSCs are clearly seen at 1 h after cell administration (area marked with solid red arrow) and no longer detected at 24 h (area marked with empty red arrow).

**Figure 7 life-13-00288-f007:**
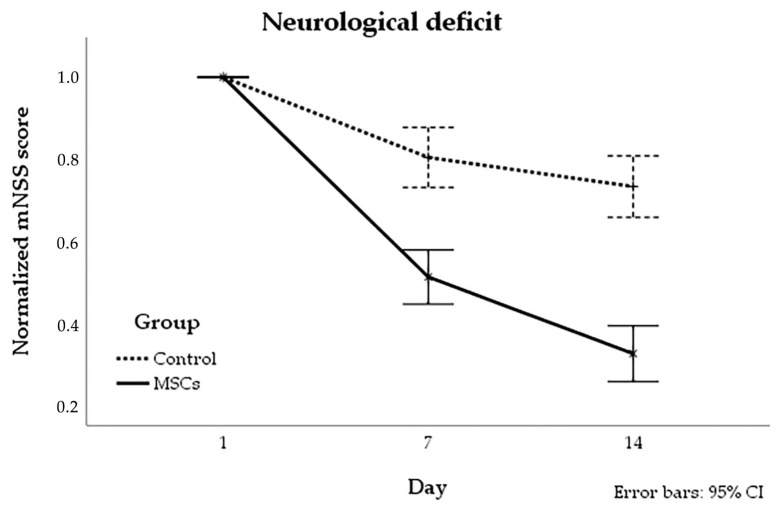
The dynamics of changes of the neurological deficit in experimental groups. Neurological deficit was assessed using the mNSS before and on the 7th and 14th day after the IV injection of MSCs (solid line) and saline (dotted line). The dynamics of changes of the normalized mNSS in the cell therapy group differ significantly from that of the control group.

**Figure 8 life-13-00288-f008:**
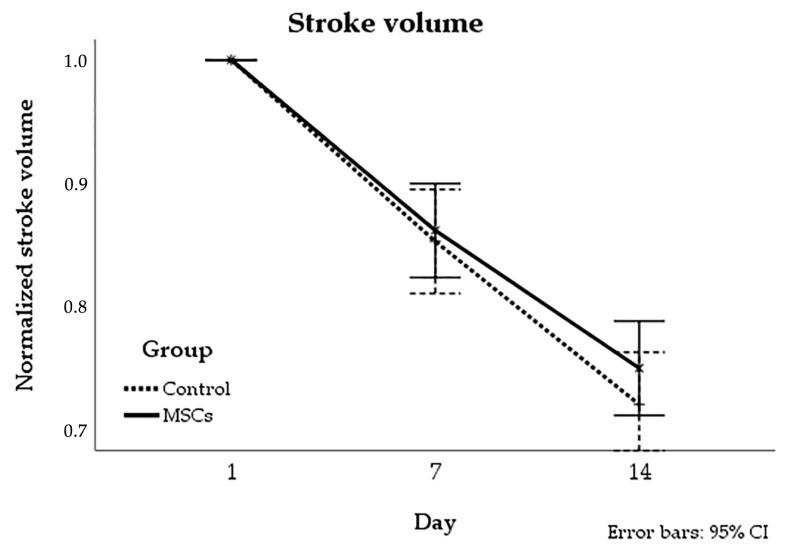
The dynamics of changes of stroke volume in experimental groups. The volume of the infarct zone was estimated using MRI performed before and on the 7th and 14th days after IV injection of MSCs or saline. No significant differences in the dynamics of stroke volume changes between the two groups were observed.

## Data Availability

The data presented and analyzed in this study can be available from the corresponding author upon reasonable request.

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
