# Peer review of "Dynamic MRI of the Mesenchymal Stem Cells Distribution during Intravenous Transplantation in a Rat Model of Ischemic Stroke"

_life, 2023, doi:10.3390/life13020288_

Round 1

Reviewer 1 Report

In the manuscript entitled “Real time MRI of the mesenchymal stem cells distribution after intravenous transplantation in experimental stroke”, Cherkashova et al. used magnetic resonance imaging (MRI) to visualize and estimate the timelapse distribution of superparamagnetic iron oxide (SPIO) labeled mesenchymal stem cells (MSCs) in live ischemic rat brain. After the transient idle cerebral artery occlusion, i.e., stroke, the MRI was performed during and after the intravenous transplantation of MSCs. Based on the MRI data, the authors evaluated the short-term engraftment of MSCs in the brain and the long-term therapeutic efficacy of the transplantation.

Overall, this study is well conducted, and the paper is well written. It is indeed novel to use high-temporal-resolution (7 min) MRI to visualize MSCs in the brain during and after the injection. The statistically significant difference in the neurological deficit between the studied and control group shows the efficacy of the approach. Although this approach cannot pinpoint the exact mechanism of the MSCs, this paper shows that MRI can be a powerful tool for studying stem cell distributions during and after administration. However, there are some issues that need to be addressed before the manuscript can be considered for publication.

1.     It is incorrect to use the term “real time” to describe the presented MRI approach. In the imaging community, “real time” normally refers to a frame rate of at least 10 Hz. However, in this study, the temporal resolution is only 7 min. Therefore, all the “real time” claims throughout the manuscript need to be changed.

2.     The MSC accumulations in the MRI images are too small to be observed clearly. The authors have used arrowheads to mark their locations, but there are also many “black spots” that look very similar to the marked spot but are not marked. The authors should (1) provide zoom-in versions of the MSC accumulations for better visibility, or (2) mark all the MSC accumulation spots with a different color.

3.     What is the spatial resolution of your MRI system? What are the sizes of MSCs? Is it really true that your MRI can resolve MSCs at the single-cell level?

4.     The authors claim that they “invented a very sensitive MRI method”. However, the MRI method has already been invented before. Also, I do not see why the presented MRI is more sensitive than state-of-the-art MRI approaches.

5.     The abbreviations in Figure 1 should be defined in the caption.

6.     All the MRI images should be provided with scale bars. 

Author Response

We would like to thank you for reviewing our manuscript and your expert opinion. We made the following amendments according to your helpful suggestions.

  1. “It is incorrect to use the term “real time” to describe the presented MRI approach. In the imaging community, “real time” normally refers to a frame rate of at least 10 Hz. However, in this study, the temporal resolution is only 7 min. Therefore, all the “real time” claims throughout the manuscript need to be changed”.

Thank you for your remark. We partly agree with it. However, in our previous study [doi.org/10.3389/fnins.2021.641970] and in the works of some authors [doi:10.1177/0271678X16665853, doi:10.1161/STROKEAHA.117.018288] the described MRI approach (visualization of cell distribution starting at the moment of transplantation) is called “real time”. Nevertheless, in order to avoid confusion, we have changed “real time MRI” to “dynamic MRI” according to your suggestion.

  1. «The MSC accumulations in the MRI images are too small to be observed clearly. The authors have used arrowheads to mark their locations, but there are also many “black spots” that look very similar to the marked spot but are not marked. The authors should (1) provide zoom-in versions of the MSC accumulations for better visibility, or (2) mark all the MSC accumulation spots with a different color».

Thank you for your comment. We would like to clarify that on the presented MR images all zones of labeled cells’ accumulation are marked with red arrows and there are really few of them. As we mentioned at the beginning of the results section, we distinguished hypointense areas that correspond to SPIO labeled cells from the other hypointense “spots” (cerebral veins, microbleeds, etc.)  by comparing the series of adjacent slices before and after transplantation in dynamics. Figure 2 presents this dynamical approach and here transplanted cells are marked with red arrows at the time of their appearance. The images are provided with the highest possible resolution with the ability to enlarge them by the reader.

  1. «What is the spatial resolution of your MRI system? What are the sizes of MSCs? Is it really true that your MRI can resolve MSCs at the single-cell level?»

High resolution Susceptibility Weighted Imaging has resolution of 0.1x0.1x0.5 mm. On MRI we visualize the disturbance of the local magnetic field created by the SPIO label, which appears as hypointense (“black”) spots on T2*WI and SWI. The size of these spots is usually much bigger than the real size of the single cells. The possibility to visualize single SPIO labeled cell on MRI was demonstrated in several studies [10.2147/IJN.S101141, 10.1016/S0006-3495(99)77182-1, 10.1002/mrm.20718, 10.1002/mrm.20747], as well as in our previous work [10.1371/journal.pone.0186717]. We added this information to the introduction section.

  1. «The authors claim that they “invented a very sensitive MRI method”. However, the MRI method has already been invented before. Also, I do not see why the presented MRI is more sensitive than state-of-the-art MRI approaches».

The originality of the present research lies in the application of the sensitive dynamic magnetic resonance imaging method for visualization of stem cells directly in the process of intravenous transplantation. We agree with your remark and rephrased the sentence in question.

  1. «The abbreviations in Figure 1 should be defined in the caption».

The correction was made.

  1. «All the MRI images should be provided with scale bars».

Unlike histological images, MR images are commonly presented without scale bars.

Reviewer 2 Report

The manuscript entitled "Real time MRI of the mesenchymal stem cells distribution after 2 intravenous transplantation in experimental stroke" describes a study utilizing susceptibility-weighted imaging (SWI) technique to monitor the distribution of transplanted mesenchymal stem cells (MSCs) in a rat brain with experimental strok model. The results demonstrated that real time SWI was able to detect the tansplanted MSCs in the rat brain sequentially, and the nerological deficits were decreased after injecting the MSCs intraveenously.

Comments:

1) Title should include "rat brain" or "animal model".

2) Lines 29. This study did not invent an new MRI technique. Actually, authors only utilized a real time SWI scanning protocol. 

3) Line 102. T2*wi should be corrected to T2*WI.

4) Line 159 and 161. DPBS and FBS should be spelled out when firstly used. All the acronyms used in this manuscript should be spelled out when first use.

5) Line 188. T2-wi should be corrected to T2WI. DWI should be spelled out.

6) Line 197, figure 1. All the abbreviations should be spelled out in the figure legend. T2wi should be corrected to T2WI, and "Hystology" should be corrected. In addition, a detailed description for the figure should be written in the figure legend.

7) Line 226. What is "the control MRI"?

8) Line 234. "susceptibility weighted images" has been changed to SWI.

9) Line 249. "perfume the complete assessment...." I don't understand this sentence, maybe there is a typo here.

10) Line 259. The "DWI" has been defined earlier.

11) Line 267. The "SWI" has been defined earlier.

12) Lines 270-271. A temporal resolution of 7 min 15 sec and 10 repeats should result in 702 min 30 sec. It is impossible to complete the MRI scan within an hour. 

13) Lines 278-283. What image was used to estimte stroke volume and how to define the infarct zone? If the areas were defined manually, how did authors deal with the manual errors.

14) Line 304. Is the P-value corrected? If yes, what method was used?

15) Line 306. Again, the SWI has been defined.

16) Line 354. T2w image should be corrected to T2WI. Similar problems should be corrected throughout the manuscript.

Author Response

Thank you very much for reviewing our manuscript. We highly appreciate your comments and tried to revise the manuscript according to your suggestions.

1) «Title should include "rat brain" or "animal model"».

Thank you for the remark. We consider it reasonable to add this additional information to the title as it may help to provide more accurate scientific search. The title was changed.

2) «Lines 29. This study did not invent an new MRI technique. Actually, authors only utilized a real time SWI scanning protocol».

The originality of the present research lies in the application of the sensitive dynamic magnetic resonance imaging method for visualization of stem cells directly in the process of intravenous transplantation. We agree with your remark and rephrased the sentence.

3) «Line 102. T2*wi should be corrected to T2*WI».

Thank you for the remark. The correction was made.

4) «Line 159 and 161. DPBS and FBS should be spelled out when firstly used. All the acronyms used in this manuscript should be spelled out when first use.»

The corrections were made.

5) «Line 188. T2-wi should be corrected to T2WI. DWI should be spelled out».

The corrections were made.

6) «Line 197, figure 1. All the abbreviations should be spelled out in the figure legend. T2wi should be corrected to T2WI, and "Hystology" should be corrected. In addition, a detailed description for the figure should be written in the figure legend.»

Thank you for your remark. We added the required descriptions in the figure legend.

7) «Line 226. What is "the control MRI"?»

We named the MRI study that was performed 24 hours after stroke modeling “the control MRI”. However, to avoid misunderstanding we have removed it and added a more accurate explanation.

8) «Line 234. "susceptibility weighted images" has been changed to SWI».

The correction was made.

9) «Line 249. "perfume the complete assessment...." I don't understand this sentence, maybe there is a typo here.»

The typo was corrected.

10) «Line 259. The "DWI" has been defined earlier.»

Yes, it has been, but we decided to repeat it in the Materials and Methods section.

11) «Line 267. The "SWI" has been defined earlier.»

Yes, it has been, but we decided to repeat it in the Materials and Methods section.

12) «Lines 270-271. A temporal resolution of 7 min 15 sec and 10 repeats should result in 702 min 30 sec. It is impossible to complete the MRI scan within an hour.»

Thank you very much for finding this typo. We made 8 (!) repeats of high resolution SWI and therefore completed MRI study in 58 minutes (approximately 1 hour). We made corrections throughout the text and figures.

13) «Lines 278-283. What image was used to estimate stroke volume and how to define the infarct zone? If the areas were defined manually, how did authors deal with the manual errors.»

Сalculation of stroke volume T2WI and manual segmentation was performed as described in the Materials and Methods section. For all images the segmentation of the infarct zone was performed by the same person (by a radiologist) to reduce intra-investigations error.

14)» Line 304. Is the P-value corrected? If yes, what method was used?»

General Linear Model with repeated measurements was used, as it’s mentioned above in line 302.

15) «Line 306. Again, the SWI has been defined.»

The correction was made.

16) «Line 354. T2w image should be corrected to T2WI. Similar problems should be corrected throughout the manuscript.»

The corrections were made.

Round 2

Reviewer 1 Report

The authors have not sufficiently addressed my comments.

First, regarding the presentation of the MSC accumulations in the MRI images, those accumulations are difficult to visualize, and I'm not convinced why the other "black dots" presented in the images are NOT MSC accumulations. Clearly, not all the "black dots" were labeled with red arrows. Also, I do not understand why the authors could not provide zoom-in versions of the MSC accumulations. The image quality is terrible even if I try to enlarge them in my pdf reader.

Second, I still cannot see the information in the manuscript about the spatial resolution of the MRI system and the sizes of MSCs.

Author Response

We have made the following corrections according to your suggestions, please find it below:

1) First, regarding the presentation of the MSC accumulations in the MRI images, those accumulations are difficult to visualize, and I'm not convinced why the other "black dots" presented in the images are NOT MSC accumulations. Clearly, not all the "black dots" were labeled with red arrows. Also, I do not understand why the authors could not provide zoom-in versions of the MSC accumulations. The image quality is terrible even if I try to enlarge them in my pdf reader.

We have changed the fig.2: add zoom-in pictures + marked zone of hypointense signal («black spots») with red arrows (labeled MSCs accumulation) and blue arrows (cerebral veins). We would like clarify that some cerebral veins are also hypointense (dark) on SWI, because deoxyhemoglobin is also a strong paramagnetic and can produce the disturbance of the local magnetic field to which SWI is highly sensitive [https://doi.org/10.53347/rID-13858]. In attempt to improve SPIO labeled cells visualization we use pure oxygen for the anesthesia as we described in results section («To improve the quality of cells’ visualization and for better distinction between the images of cells and cerebral vessels the mixture of isoflurane and oxygen instead of isoflurane and air was used for inhalation anesthesia. This approach provides an increase of oxyhemoglobin and decrease of deoxyhemoglobin concentration in the venous blood. Since oxyhemoglobin is a diamagnetic and deoxyhemoglobin a paramagnetic [60], the accumulation of oxyhemoglobin in the blood reduces the visibility of cerebral veins, as shown in Figure 3.»). Moreover, we distinguished zones of labeled cells’ accumulation from cerebral veins by analyzing their shape on the series of adjustment slices and time of their appearance. It is difficult to identify cells without evaluation of images in dynamics. We added additional description in the results section to clarify this issue.

 2)     Second, I still cannot see the information in the manuscript about the spatial resolution of the MRI system and the sizes of MSCs.

We added the information about spatial resolution in the materials and methods section and the explanation about MSCs size and size of hypointense zone in results section.

Reviewer 2 Report

In statistcs, authors utilized general linear model with repeated measurements to understand the dynamic changes between the two groups, and the p-value ony indicated the significant difference of dynamic changes. However, it remains unclear whether the differences were significant at each time point. In figure 7, for example, the overall p-value cannot indicate different difference at day 7 and 14. Therefore, a two-sample test with Bonferroni correction is needed.

A correlation analysis between the number of transplanted MSCs and neurological scores may help understand the relationshiop between the MSC transplantation and neurological status. The results may further strengthen the significance of this study.

Author Response

Thank you for your recommendation. Please find our corrections and answers below:

1) In statistcs, authors utilized general linear model with repeated measurements to understand the dynamic changes between the two groups, and the p-value ony indicated the significant difference of dynamic changes. However, it remains unclear whether the differences were significant at each time point. In figure 7, for example, the overall p-value cannot indicate different difference at day 7 and 14. Therefore, a two-sample test with Bonferroni correction is needed.

Thank you for this comment. For estimation of the neurological deficit besides general linear model we additionally performed pairwise comparison between groups at day 7 and day 14 using Mann–Whitney U test with Bonferroni correction and received significant difference at this time points. We added this data to materials and methods section and results.

2) A correlation analysis between the number of transplanted MSCs and neurological scores may help understand the relationshiop between the MSC transplantation and neurological status. The results may further strengthen the significance of this study.

In the present study, we qualitatively assessed the presence or absence of cells in certain brain regions in dynamic. The approach you suggested is very logical and interesting. We will try to performed it in the future. We have also added this information to the discussion section.

Round 3

Reviewer 1 Report

The authors have addressed the rest of my concerns in the revision. I can now recommend the publication of this manuscript. 

Author Response

Thank you.

Best regards, Ilya Gubskiy

Reviewer 2 Report

The manuscript is revised adequately, but some minor issues need to be corrected. 

1. The term "real-time" sould be replaced by "daynamic" throughout the entire manuscript. The "dynamic" was used in the title, but "real-time" remained used in the results and discussion on Lines 340-341, and 470-515.

2. The T2*-weighted image should be replaced by T2*WI on Lines 478, 483, 509.

3. Lines 541, some references should be cited here for the statement "the previous studies".

Author Response

Thank you for the revision of our manuscript. The corrections were made according to your recommendations.

Best regards, Ilya Gubskiy